# Differences in Vegetative, Productive, and Physiological Behaviors in *Actinidia chinensis* Plants, cv. Gold 3, as A Function of Cane Type

**DOI:** 10.3390/plants14142199

**Published:** 2025-07-16

**Authors:** Gregorio Gullo, Simone Barbera, Antonino Cannizzaro, Manuel Scarano, Francesco Larocca, Valentino Branca, Antonio Dattola

**Affiliations:** Department of Agraria, Mediterranean University of Reggio Calabria, 89124 Reggio Calabria, Italy; barbera_simone@libero.it (S.B.); antoninocannizzaro@live.it (A.C.); agr.scaranomanuel@gmail.com (M.S.); francescolarocca43@gmail.com (F.L.); valentino.branca@unirc.it (V.B.); antonio.dattola@unirc.it (A.D.)

**Keywords:** fruit quality, leaf gas exchange, growth dynamics, hydraulic resistance

## Abstract

This study investigated the influence of cane diameter on vegetative, productive, and physiological behaviors in *Actinidia chinensis*, cv. Gold 3. Conducted over two years (2021–2022), the experiment compared canes with larger (HD) and smaller (LD) proximal diameters. This research focused on parameters such as shoot morphology, leaf gas exchange, fruit quality, and hydraulic resistance. The results revealed that HD canes promoted more vigorous growth, with a higher proportion of long and medium shoots, whereas LD canes resulted in shorter shoots. Additionally, the HD canes demonstrated a higher leaf area and more extensive leaf coverage, contributing to enhanced photosynthetic activity, as evidenced by enhanced gas exchange, stomatal conductance, and transpiration rates. This higher photosynthetic efficiency in HD canes resulted in more rapid fruit growth, with a larger fruit size and weight, particularly in fruits from non-terminate shoots. By contrast, fruits on LD canes exhibited slower growth, particularly in terms of fresh weight and dry matter accumulation. Despite these differences, maturation indices, including soluble solids and acidity levels, were not significantly affected by cane type. The findings suggest that selecting HD canes during winter pruning could lead to earlier harvests, with improved fruit quality and productivity, making this practice beneficial for optimizing vineyard management in *Actinidia chinensis*.

## 1. Introduction

Italy ranks second in global kiwi production [1], and the southern part of the peninsula, thanks to its mild winters, represents a significant production area of this country. The *Actinidia deliciosa*, particularly the ‘Hayward’ cultivar, which was key to the global success and expansion of kiwifruit cultivation after the second half of the 20th century, requires approximately 650 or more chilling units [2] and produces green-fleshed fruits that typically ripen around the first 10 d of November.

Several agronomic aspects related to pruning have been extensively studied in ‘Hayward’, including the type of cane [3], cane age [4], inclination [5,6], and leaf area index [6,7].

In mild-climate environments such as southern Italy, where chilling requirements are not always met, some studies have helped define the optimal bud load and the ideal cane length to retain during winter pruning, in order to achieve satisfactory yields both in terms of quantity and quality [8,9]. The absence of autumn frosts allows fruit to be harvested at a more advanced ripening stage, ensuring better post-harvest storability.

Since the early 2000s, alongside *A. deliciosa*, the cultivation of *A. chinensis* cultivars, mainly yellow-fleshed, has also spread. These cultivars are characterized by lower chilling requirements and are therefore better suited to mild-winter environments such as southern Italy. Among them, the ‘Gold 3’ cultivar has attracted particular interest: it requires fewer chilling units than ‘Hayward’, flowers approximately 15 d earlier, and produces fruit that can be harvested in the first 10 d of October [10].

The growth habitus of *Actinidia* (kiwi) exhibits an acrotonic behavior, that is, the tendency to develop vigorous shoots in the apical part of a branch compared to the basal part; therefore, shoot growth is reduced from the distal to proximal parts of the kiwifruit cane, particularly if the parental shoot is free of leads. However, as reported by several authors, shoot growth is influenced by factors such as genotype, environmental conditions, trained system, and competition established between new shoots [11,12,13,14,15,16,17]. According to [18], in the first year, axillary meristems on the parent shoot (cane) initiate a fixed number of phytomers before winter dormancy (preformed phytomers), which expand in the following season when the shoot apical meristem (SAM) initiates additional new phytomers that can expand during the current season (neoformed phytomers). As noted, kiwi buds have the potential to develop into long, non-determinate shoots [19]. The vine must support the rapid formation of a new canopy and flower development between bud break and flowering, utilizing previously stored reserves [20,21,22] until the shoot becomes autotrophic, approximately 40 d after bud break [23]. The timing of bud break, influenced by the satisfaction of cold requirements, affects the competition between early- and late-budding buds. This competition often leads to limited resources and can result in a premature abortion of the shoot apical meristem (SAM), causing underdeveloped shoots that fail to produce preformed phytomers. Interestingly, once the shoots become autotrophic, the xylem flow could play a crucial role in the development of new, non-preformed phytomers. Acrotony, in particular, favors the development of apical shoots and the bending of the cane in the proximal positions. Three types of shoots in *Actinidia* have been reported: spur, long, and short. The removal of neighboring shoots has been shown to increase the proportion of long shoots [11,24,25]. Conventional pruning studies on kiwifruit, which focus on selecting canes originating from the cordon during winter pruning and give less importance to summer pruning, have shown that in ‘Hayward’ kiwis, thicker canes tend to produce fruit with better quality traits at harvest compared to thinner ones [26]. The role of the cane is a key element in understanding and managing the productive balance of the *Actinidia* tree. Excessively vigorous canes tend to develop shoots with long internodes and fewer floral buds, while weak ones fail to provide adequate mechanical support or sufficient assimilate translocation [27].

Their proper selection during winter pruning allows for the establishment of balanced and sustainable fruiting. Therefore, it is necessary to identify, for each cultivar, the average vigor of well-lignified canes correctly exposed to light [28]. However, the effect of cane diameter on vegetative growth dynamics and on the fruit’s carpometric and organoleptic characteristics remains unclear, and few studies have been conducted in this way on *A. chinensis* species. The objective of the present study was to evaluate the influence of the diameter of renewal canes on vegetative growth dynamics, as well as on the carpological and qualitative parameters of the fruit, and to assess whether it could affect harvest timing based on the organoleptic characteristics of the final product, as a result of the differing vegetative behavior between the two cane types.

The present study aimed to explore the dynamics of shoot development in *Actinidia*, focusing on both vegetative and productive aspects. The canes differ from one another in terms of diameter. The influence of various factors, such as shoot morphology and fruit quality, was observed, taking into account the differences both during the first growth phase in the heterotrophic stage and later in the autotrophic stage.

## 2. Materials and Methods

### 2.1. Experimental Site and Plant Material

The experimental trial was conducted over two years (2021/2022) in a six-year-old *Actinidia* orchard spanning approximately 2.5 hectares, located in southern Italy, on the *A. chinensis* Planch, cultivar Gold 3. The plants were grafted onto Hayward rootstock. Two pollinator cultivars, M33 and M91, were used for Gold 3, with a male-to-female plant ratio of 1:6. The plants were trained in a pergola system, with an in-row spacing of 4 and 5 m between rows (500 plants per hectare).

The average maximum temperature was reached in July and August (Appendix A). The precipitation was mainly concentrated in the autumn–winter period (Appendix A). The chilling units, calculated using the Utah model, were 469 for 2021 and 458 C.U. for 2022. These values are in line with the chilling requirements of the cultivar of *A. chinensis* [29], ensuring proper dormancy break and plant development. The orchard was equipped with a sprinkler irrigation system delivering 8 L h^−1^ plant^−1^. The irrigation season extended from late April to late October, with daily irrigation sessions lasting approximately 8 h, resulting in a total water volume of 60 L per plant per session. During winter pruning, plants were standardized in terms of the number of canes (16 canes per plant), cane length (25 buds per cane), and total buds per plant (400 buds per plant).

### 2.2. Experimental Design

A randomized block design was established with three blocks, each consisting of 20 continuous linear meters of the pergola canopy, containing six plants. The experimental plan was prepared during the winter. During winter pruning, canes with a diameter of less than 15 mm were removed. Eight canes with diameters ranging from 1.4 to 1.6 cm and eight canes ranging from 1.9 to 2.1 cm were retained.

Two treatments were applied within each block following the method reported by [26] for another kiwifruit species:‒HD cane with a larger proximal diameter (~2 cm);‒HL cane with a smaller proximal diameter (~1.5 cm).

Eighteen canes per treatment were selected for each block (3 canes per plant), totaling 54 canes per treatment: 2 canes per plant were used for all observations on the plant and 1 cane per plant was used for fruit collection and laboratory analysis. In total, we observed 36 canes per treatment per block for monitoring and 18 canes for treatments per destructive analysis.

In particular, 12 canes per treatment per block (total 36 canes) were used to monitor vegetative development and fruit diametrical growth, while 6 canes per treatment and per block (a total of 18 canes) were used to carry out detached fruit to destructive analysis in the pre-harvest observation epochs: 6 fruits were taken in a random manner from these canes (1 fruit per cane) for a total of 18 fruits per treatment for each sampling epoch (4 dates).

### 2.3. Field and Laboratory Measurements

From the first 10 d of April, during vegetative growth, the phenological stages of each cane were monitored using the BBCH scale. Data were collected from the two canes per treatment and per plant under observation, identifying fertile shoots, vegetative shoots, and closed buds. At flowering, the number of flowers per fertile shoot was recorded. Biweekly measurements were adopted according to [22] based on the number of leaves and the corresponding length in centimeters:

Spur (SP): <10 leaves ≤ 15 cm;

Medium (MS): 10–18 Leaves = 16–60 cm;

Long (LS): >18 leaves > 60 cm.

The percentage distribution of each shoot type (SP, MS, LS) was calculated for each cane type (LD and HD). From early July until harvest, 18 fruits per cane type were randomly sampled biweekly from the additional canes selected for each treatment (3 fruits per cane type per plant). For each time period, one cane from each of the six plants per block was used for data collection. In the laboratory, the fruits underwent detailed carpometric analysis, and the dry matter content was measured. From late August (73 DAFB), additional ripening indices were determined, including flesh color, dry matter, total soluble solids (TSSs), titratable acidity (TA), and the TSS/TA ratio.

### 2.4. Physiological Measurements

#### 2.4.1. Gas Exchange

The net photosynthesis (Pn), transpiration (Tr), and intercellular CO_2_ concentration (Ci) were measured on twelve leaves per treatment per block. The measurements were taken using a portable photosynthesis system (Li-Cor 6400XT; LI-COR Biosciences, Lincoln, NE, USA). The gas exchange measurements were carried out during clear, sunny summer days. Data collection occurred between 11:00 and 13:00 daylight saving time (equivalent to 10:00–12:00 standard time) in order to assess the performance of the two cane types under conditions of highest daily stress. Measurements were repeated during the last week of the summer months (June, July, and August) in both years [30].

#### 2.4.2. Cane Hydraulic Conductance

For this purpose, a Hydraulic Conductance Flow Meter XP Gen3 (Dynamax Inc., Houston, TX, USA) was used. The canes were removed from the cordon with a clean cut and connected to the HCFM. Shoot hydraulic conductance (HC) was measured in quasi-steady mode at a pressure of 0.3 MPa. The sample was perfused with high-pressure water until the leaves were visibly waterlogged. HC was monitored once this hydration state was reached until it became stable, and a quasi-steady state (QSS) reading was taken [31].

#### 2.4.3. Chlorophyll Fluorescence Parameter

Chlorophyll fluorescence parameter data were obtained using a chlorophyll fluorimeter (LI-COR 6400-40; LI-COR Biosciences) from clips of the same mature leaves selected for gas exchange measurements after 30 min of preconditioning in the dark. The following chlorophyll fluorescence parameters were determined: F_0_ minimum fluorescence and Fm maximum fluorescence. These values were subtracted and divided [(Fm − F_0_)/F_m_] to calculate the maximum quantum efficiency of photosystem II (PSII) photochemistry (F_v_/F_m_), which provides quantitative information on the current state of plant performance under stress conditions. The non-photochemical quenching coefficient (NPQ) was also measured.

### 2.5. Morphometric and Maturation Indexes of the Fruit

#### 2.5.1. Fresh Weight

The fresh fruit weight (FW) was measured using an analytical balance (Precisa, BJ 610C, Dietikon, Switzerland). Polar and equatorial diameters were measured with a precision digital caliper. The transversal diameter was calculated using the mean value of the two transverse diameters (L) and their ratio. The relative length was calculated as the ratio of the polar diameter (H) to the mean value of the equatorial diameter (L).

#### 2.5.2. Colorimetric Analysis

Flesh color was determined using the CIELab color space, adopted by the International Commission on Illumination (CIE). This method quantified the color of the pulp and flavedo based on three coordinates: L* (lightness), a* (red–green chromaticity), and b* (yellow–blue chromaticity). Measurements were performed using a Minolta CM-700d tristimulus colorimeter with an 8 mm target mask. Prior to measurements, the device was calibrated with a white reference plate. Data were collected at four equidistant points along the equatorial zone of each fruit under the standard D65 illuminant (daylight, 6504K) and a 10° observation angle. Color parameters included brightness (L*), chromaticity coordinates (a*, b*), chroma (C*), and hue angle (h°), which were calculated as follows:

C*: $\sqrt{a^{2}}+b^{2}$

h°: arctan $\frac{b}{a}$

#### 2.5.3. Dry Matter Content

The dry matter content was determined from a pulp slice dried in an oven (Binder EED240, Tuttlingen, Germany) at 70 °C until a constant weight was reached. Dry matter was expressed as a percentage using the formula: (DW/FW) × 100.

#### 2.5.4. Total Soluble Solids (TSSs)

The TSS content was measured with a digitally temperature-compensated refractometer (Atago PAL-1, Tokyo, Japan). The sugar content, expressed in °Brix, was measured from filtered juice obtained by manually pressing the entire fruit. A mean of 36 measurements per treatment was used.

#### 2.5.5. Titratable Acidity (TA)

Titratable acidity (TA) was determined by potentiometric titration (Hach, TitraLab AT1000 Series, Loveland, CO, USA) of the juice with 0.1 N NaOH beyond pH 8.1 according to the AOAC method, with the results expressed as g L^−1^ of citric acid equivalent.

#### 2.5.6. Nutraceutical Parameters

##### Extraction of the Matrix

Ten gs of pulp from the different treatments compared was mixed with 50 mL of an extraction solution composed of ethanol, demineralized water, and acetic acid (80:19:1) and homogenized using a digital Ultra-Turrax T25 homogenizer (Janke and Kunkel, IKA Labortechnik, Staufen, Germany) at 20,000 rpm for 60 s. The homogenate was left to macerate for 48 h at +4 °C. The extract was then transferred to 50 mL Falcon-type test tubes and centrifuged at 4000× rpm for 20 min at +4 °C. The supernatant was transferred into 1.5 mL Eppendorf tubes and stored at −20 °C. The supernatant samples were subsequently used as the matrix for the following nutraceutical analyses: Total Polyphenols (TPs), Total Antioxidant Capacity (TAC), and Total Flavonoids (TFs).

##### Total Polyphenol Content

The Total Polyphenol (TP) content was determined using the Folin–Ciocalteu method, following the protocol described by [32]. This method, commonly employed for phenolic compound determination, utilized the same extraction procedure described for antioxidant capacity determination. The Folin–Ciocalteu assay is based on the chemical oxidation of phenolic compounds by an oxidizing mixture known as the Folin reagent, composed of phosphotungstic acid (H_3_PW_12_O_40_) and phosphomolybdic acid (H_3_PMo_12_O_40_). Upon reduction, it forms a mixture of tungsten and molybdenum oxides (W_8_O_23_ and Mo_8_O_23_) with a blue color. A 50 µL aliquot of the supernatant was mixed with 2.5 mL of Folin–Ciocalteu reagent (1:10), followed by the addition of 450 µL of demineralized water and 2 mL of 7% (*w*/*v*) NaCO_3_ within 8 min. The reaction mixture was incubated for 120 min at 20 °C. After incubation, the optical density (absorbance) was measured using a double-beam UV/Vis spectrophotometer (Lambda 35, Perkin Elmer Corporation, Springfield, IL, USA) at a wavelength of 765 nm in a 1 cm path-length cuvette against a blank prepared with the extraction solution alone. A calibration curve was constructed by preparing a gallic acid stock solution at a concentration of 5 g/L in 10% ethanol. From this stock solution, 1, 2, 3, 5, 10, and 15 mL were transferred into six 100 mL flasks and diluted to the volume with distilled water. Absorbance values were interpolated with known gallic acid concentrations. The total phenol content is expressed as mg gallic acid/100 g of fresh weight.

##### Total Antioxidant Capacity

The Total Antioxidant Capacity (TAC) was determined using the ABTS assay [33], an analytical method employing spectrophotometric measurements to assess the antioxidant capacity of a sample. The absorbance of a solution containing the ABTS•+ radical, generated by the oxidation of ABTS (2,2′-azinobis (3-ethylbenzothiazoline-6-sulfonate)), was measured. In its radical form, ABTS•+ is colored and absorbs in the visible range. The addition of antioxidant molecules, capable of donating hydrogen or electrons, reduces the radical to its colorless form, resulting in a discoloration proportional to the antioxidant concentration. This discoloration is measured as a decrease in absorbance over time at a specific wavelength (734 nm). The antioxidant capacity is expressed relative to absorbance values obtained for known quantities of Trolox (1.5, 1, 0.5, and 0.25 mM). A Trolox stock solution was prepared by dissolving 0.0313 g of Trolox in H_2_O in a 25 mL volumetric flask to yield a 5 mM solution. This method is simple, rapid, and allows the measurement of both hydrophilic and lipophilic antioxidants across a wide pH range. The total antioxidant content is expressed as µmol Trolox/g fresh weight (FW).

##### Total Flavonoid Content (TF)

The Total Flavonoid (TF) content was determined using a colorimetric method described by [34]. Briefly, 0.25 mL of hydrophilic extract or standard quercetin solution was added to 1.25 mL of distilled water. Subsequently, 75 µL of 5% sodium nitrite (NaNO_2_) solution was added. After 6 min, 150 µL of 10% aluminum chloride (AlCl_3_) solution was added, and after an additional 5 min, 0.5 mL of 1 M sodium hydroxide (NaOH) was added. The mixture was brought to a final volume of 2.5 mL with distilled water and thoroughly mixed. The absorbance was promptly measured using a double-beam UV/Vis spectrophotometer (Lambda 35, Perkin Elmer Corporation, USA) at a wavelength of 510 nm in a 1 cm path-length cuvette against a blank prepared with the extraction solution alone. The results are expressed as mean (±SE) mg quercetin equivalents per g of fresh weight.

#### 2.5.7. Harvest and Pruning Residues

At harvest, the number of fruits and the production weight were determined for each shoot type and each cane. The yield per cane was separated into 5 size classes. After harvest, the selected canes from each plant were manually defoliated, and the number of leaves and total leaf area for each shoot type and each cane were measured using a Licor 3100 area meter. In January of both years, the canes were removed, weighed, and then placed in an oven at 105 °C until a constant dry weight was achieved.

### 2.6. Statistical Analysis

Statistical data analysis was performed using SPSS software v. 22.0 (IBM Corporation, New York, NY, USA), while graphical processing was conducted with SigmaPlot v. 10.0 (Systat Software Inc., San Jose, CA, USA). Data were statistically analyzed using a two-way ANOVA or a *t*-test.

Data were statistically analyzed using a two-, three-, or four-way ANOVA, applying a t-test when the number of levels of the independent variables was 2 and Tukey’s test when the number of levels exceeded 2.

## 3. Results

The diameters of the two selected canes for each treatment were significantly different; indeed, in both years, the high-diameter (HD) cane had an average diameter (23.10 mm ± 0.34 in 2021; 23.48 mm ± 0.49 in 2022), which was 35% larger than the average diameter of the low-diameter (LD) canes (15.59 mm ± 0.29 in 2021; 16.11 mm ± 0.33 in 2022). During the first 10 d of April, the percentage of buds in the closed stage (CS) was 13% in the LD canes, while it was higher in the HD canes, at about 19%. However, the difference was not significant (Table 1). No significant variation was also observed regarding the percentage of the buds in the budding stage (BS; stage BBCH from 1 to 10) between the two cane types. Finally, the percentage of buds in the phenological vegetative stage (VS; stage BBCH from 10 to 18) was higher in the LD canes (31.57%) than in the HD canes, but again, the differences were not significant (Table 1). Analyzing the evolution of the phenological stages in the subsequent observation periods, it is evident that there was no difference between the two types during the second 10 d of April and the first 10 d of May. In both cane types, the evolution of the phenology stage from the BS to the VS stage was evident across the three observation dates, while the percentage of CS buds remained unchanged over the three observation periods. The values of the above parameters were generally higher in the first year, but the difference was not significant, and no Y × T interaction was observed.

The smaller-diameter cane (LD) was characterized by a higher susceptibility to short shoot sprouting; indeed, the percentage of short shoots (SP) was significantly higher (by 33%) in the LD canes compared to the alternative canes (HD), while the percentage of medium shoots (MSs) was 9% higher in the HD compared to the LD canes (Table 2). Finally, the percentage of longer shoots (LSs) was significantly higher in the HD canes (Table 2). No significant changes were recorded over the two years of observation, and no Y × T interaction was detached for these parameters (Table 2). The growth pattern of short shoots did not differ according to cane size.

The SP reached their maximum length in both cane types during the second 10 d of May (14 DAFB) (Figure 1), 9.6 cm ±0.3 in LD and 10.1 cm ± 0.4 in HD, without significant differences. The average length value was always higher in the larger-diameter shoots; however, the statistical difference between the spur shoots of the two cane types was not significant (Figure 1) at any data point. The development of the MSs length showed substantial differences between the two cane types. In fact, the length of the MSs was significantly greater in the HD canes (53 cm 2021; 56 cm 2022) compared to the LD canes (43 cm 2021; 45 cm 2022) until 40 DAFB (the first 10 d of June) (Figure 2).

In particular, the growth of MSs stopped in the HD canes (Figure 2), whereas it continued slowly until 60 DAFB (the last 10 d of June) in the LD canes. Therefore, the differences disappeared by 60 DAFB; indeed, the MSs reached an average value that was similar in both cane types (Figure 2). The length development of the LSs was always significantly different between the two cane types, with the length being higher in the larger-diameter canes (HD), reaching a final length of 200 cm, while in the LD canes, the average length was about 130 cm. This behavior was consistent for each shoot type in both years, and no interaction between treatment (T) and year (Y) was observed (Figure 3). The leaf parameter values were statistically influenced by the type of cane.

Specifically, for the spur and terminate shoots, the average number of leaves and their leaf area were similar across the two treatments (Table 3). Conversely, the leaf parameters differed between the two treatments for non-terminate shoots; indeed, the number of leaves and leaf area were significant and 50% higher in the HD canes compared to the LD canes (Table 3). More specifically, the contribution to the total leaf area in the HD canes was provided by the long shoots (59%), followed by the medium shoots (30%), and finally the short shoots (11%). By contrast, in the LD canes, short shoots (SP) contributed 50% to the total leaf area, the medium shoots (MSs) contributed 35%, and the long shoots (LSs) contributed 15%. Finally, in the LD canes, the average leaf area was 1.41 m^2^ (±0.11), whereas in the HD canes, the leaf area was 51% higher, exceeding two square meters (2.14 m^2^ ± 0.15) (Figure 4). No significant differences between years and no year × treatment interaction were observed.

With regard to gas exchange, significant differences were observed in Pn (net photosynthesis), stomatal conductance (gs), transpiration (Tr), and Ci. With regard to the month of June, net photosynthesis (Pn) was significantly higher in the HD canes, recording values of 18.40 µmol CO_2_ m^−2^ s^−1^, which were significantly greater than those reported in the LD canes, where values remained below 17 µmol CO_2_ m^−2^ s^−1^. Stomatal conductance was also significantly higher in the HD canes compared to the LD canes, with values of 0.18 µmol H_2_O m^−2^ s^−1^ and 0.22 µmol H_2_O m^−2^ s^−1^, respectively. A similar trend was observed for transpiration (E), with significantly higher values in the HD canes (2.94 mmol H_2_O m^−2^ s^−1^ ± 0.88) compared to the LD canes (2.22 mmol H_2_O m^−2^ s^−1^ ± 0.88). Among the months, the values were lowest in August for almost all parameters in both HD and LD canes. However, in the latter case, the values recorded in June were very similar to those in July. These values were higher in the second year, while no significant treatment × year (TxY) interaction was observed for almost all parameters.

In terms of fluorescence parameters, the quantum efficiency ratio was consistently below the optimal value (0.85) [35] but was significantly higher in the HD treatment. Similarly, PhiPS2 and ETR showed significant differences, being higher in the HD canes. Regarding average leaf temperature, the value was consistently more than 0.5 °C lower in the leaves of the HD treatment. In the second year, all values were significantly different, while no interaction TXY was recorded. The leaf water potential was higher in the HD canes. Statistically significant differences were observed between treatments in hydraulic conductivity (Table 4). In fact, the HD shoot showed values of 5.781 × 10^−4^ kg/s Mpa, while the LD treatment, in comparison, had values close to 1.9 × 10^−4^ kg/s MPa. However, no significant difference was observed between years, and no treatment × year interaction was observed. The number of fruits produced was greater in the cane with a larger diameter, both in terms of fruit number and weight (Figure 5). However, no variations were observed when considering the number of fruits per shoot type in relation to cane diameter.

A significantly lower number was observed in the SP shoots compared to the MSs and LSs, while no difference was found between the MSs and the LSs (Table 5).

It is clear that the different presence of short, medium, and long shoots on each type of cane changed the fruit load for the cane (Figure 5). Regarding fruit size classes, 42% of the fruits in the treatment fell into the >110 g category, whereas this percentage dropped to 22% in the alternative treatment. For fruits under 110 g, a percentage greater than 22% was observed in the LD treatment compared to the HD treatment (Table 6). In the short shoots, the leaf-to-fruit ratio was significantly lower compared to both medium and long shoots, which had ratios exceeding five and nine leaves per fruit, respectively. The ratio of nine leaves per fruit in the long shoots was statistically higher than that in the other shoot types (Table 7).

Fruit growth reached saturation earlier on the HD canes compared to the LD canes (Figure 6a–c).

The fresh weight of the fruit remained similar until the second 10 d of July (73 GDPF). By the third 10 d of August (114 GDPF), the fruit reached a higher size and weight in the HD treatment compared to the cane of lower diameter (LD). Furthermore, in the HD canes, these parameters remained unchanged in the subsequent sampling dates, whereas in the LD treatment, the parameters continued to increase until the first 10 d of October, when the differences between the two treatments disappeared (Figure 6). Regarding dry weight, the fruit reached a total dry matter content of 18.68 g at the end of August in the LD canes, whereas in the alternative canes (HD), the dry weight was statistically higher, exceeding 23 g per fruit (Figure 7). This trend persisted in the subsequent sampling dates, and the values for the two cane types overlapped only in the first 10 d of October. When analyzing the dry matter content, similar values (without significant differences) were found between the two treatments at each date until harvest (Figure 8). Therefore, the overall production of the cane was different between the two types of canes analyzed (Figure 9).

Because no differences were observed in sprouting (number of fertility shoots, number of flowers per shoot), the different yield per cane was attributable to slow fruit growth in the LD canes compared to the HD canes. Between the end of September and the first 10 d of October, there were no changes in production for the HD canes, whereas for the LD canes, there was a significant increase between the two dates; furthermore, during the third 10 d of September, the calculated yield per cane was significantly lower compared to the yield per cane calculated for the same data for the alternative canes, whereas the differences between the two calculated productions were nullified in the first 10 d of October. The TSS increased from 73 to 155 DAFB, while the firmness decreased. For both parameters, no differences between treatments were observed throughout this period (Table 8). The titratable acidity and TSS-to-TA ratio differences between treatments were also not significant (Table 8). For all parameters, a significant difference between the two observation years and a non-significant y × t interaction were observed. There was an 18% increase in lightness between 70 and 110 GDPF; it remained stable at the subsequent sampling dates. By contrast, there was an increase in the a* parameter (from −6 at 73 GDPF to almost 0 at 155 GDPF), while the b* parameter gradually decreased (from 254 at 73 GDPF to 16 at 155 GDPF) during the same period (Table 9). The chroma also gradually decreased, stabilizing between 130 and 155 GDPF, while the °hue decreased to 101, 94, and 92 at 112, 133, and 155 GDPF, respectively (Table 9). However, no significant differences were observed between treatments for all colorimetric parameters at 70, 110, 130, and 155 GDPF (Table 9). It was found that TAC, TP, and TF were significantly higher in the fruit of the LD canes compared to the HD canes, while no significant changes were observed in relation to TF (Figure 10). For the nutraceutical parameters measured at harvest, TAC, TP, and TF were significantly higher in the fruits of the CH treatment than in the shoots of the BH treatment (Figure 10). Significant differences were found with respect to average fresh weight and average dry weight for each type of shoot (Figure 11).

## 4. Discussion

The type of cane did not affect the percentage of closed buds or the fertility of the shoot, contrary to the findings of [26], although their study focused on a different species. Larger canes exhibited a clear contraction or abortion of the growth point (SAM), which inhibited the expansion of preformed phytomers [36]. This condition led to a lower presence of spur shoots and a greater shoot length and surface area of non-determinate shoots. The neoformation and expansion of phytomers, as reported by [18], were favored in large canes (HD), along with a higher number of non-determinate shoots compared to the canes with smaller diameters. The experiment clearly shows that access to reserves from the parent cane is undoubtedly a factor influencing the evolution and development of preformed phytomers in the SAM, and that the amount of reserves and the fate of the bud are correlated with the diameter of the cane at the site of the developing bud, as reported by [37,38,39]. Furthermore, this situation favors the development of spur shoots [38,40], which were found to be very high in the minor-diameter reeds. However, the role of xylem flow should not be overlooked, as stated in our study. Adequate xylem flow is essential for the transport of water and nutrients to plant cells and for reducing competition between nearby shoots, thus positively influencing their growth and development, as reported in previous studies [41,42,43]. In fact, shoots that expand slowly have shorter internodes and cease growing early, while shoots that expand rapidly have longer internodes and grow for longer periods [44,45]. This effect is particularly evident when the shoot developing from the bud becomes photosynthetically active and autotrophic, as confirmed by the greater presence of both determinate and indeterminate growing shoots on the thicker cane, with the predominance of the latter, along with higher photosynthesis values. It is reasonable to hypothesize that this combination not only prevented SAM abortion but also led to the formation of new phytomers, as stated by some authors [46]. The higher conductivity of the HD canes ensured a higher water potential (less negative) and better gas exchange, as evidenced not only by photosynthetic activity but also by higher stomatal conductance and increased transpiration. The higher conductance of the HD canes can be attributed to higher xylem conductive area.

The greater leaf transpiration also lowered leaf temperature, which, during summer days, is often exposed to PPFD values exceeding 2000 µmol m^2^ s^−1^ at the experimental site, putting it at risk of photoinhibition [47,48,49]. In fact, fluorescence parameters revealed higher quantum efficiency values in the leaves of the HD canes, although the values in both cases were still far from the optimal value of 0.85 [35,50]. Therefore, the leaf of the HD canes showed a higher leaf water potential, closer to 0, compared to the other treatment. This confirmed that the leaves on the HD canes were less stressed compared to the other treatment as a consequence of the greater water flow in the HD canes.

The higher transpiration rate in the HD shoots resulted in a decrease in leaf temperature, thereby reducing the risk of thermal-induced fluorescence (Fv/Fm). This outcome, combined with increased stomatal conductance, may be responsible for the observed increase in net photosynthesis (Pn), which also affected fruit characteristics.

Regarding the effect of cane type on fruit biometric parameters, differences were observed during the growth phase, but these were nullified in terms of fresh weight by the second 10 d of September. Indeed, on the HD canes, the fruit reached the phase of maximum cell expansion significantly earlier compared to the alternative treatment, as evidenced by the evolution of its diameter, which remained unchanged from the end of August, while on the smaller cane, this process was slower and was completed in the second 10 d of September. The balance between incoming and outgoing fluxes from the fruit is responsible for the size variations [51,52], depending on whether the flux is positive or negative, respectively [53]. Fruit transpiration is very high at the beginning of the season and decreases progressively with fruit development [54], but during the fruit growth and cell expansion phase in our experiment, the higher hydraulic conductance of xylem flow in the HD canes was likely responsible for the greater fruit growth rate and the saturation of its cell growth, which was completed earlier compared to the alternative treatment. Subsequently, the role of phloem flow became significant, leading to earlier and more intense sugar loading into the fruit in the treatment with the larger cane [55,56]. In this regard, differences between the fruits of the two treatments emerged when considering the amount of dry matter accumulated per fruit, which was greater in the HD treatment. Indeed, the higher water content in the fruit of the HD treatment, combined with the higher sugar content, resulted in a dry matter equal to that observed in the alternative treatment.

## 5. Conclusions

In the experiment conducted, cane diameter influenced the vegetative and productive parameters, while no effects were observed on the percentage of closed buds or the development of phenological stages. A strong effect of cane diameter on bud type was recorded, with a predominance of spur-type shoot on low-diameter canes and long shoot on high-diameter canes. Additionally, greater growth was observed in larger-diameter canes compared to medium- and long-diameter canes. No variations were observed in the fruit/bud ratio based on bud type, while an increase in the number of leaves per fruit was noted, moving from spur-type buds to medium-length buds and reaching the maximum value in indeterminate-growing buds. The different conductivity of the canes altered photosynthetic efficiency and the fruit growth pattern, which, along with a greater leaf-to-fruit ratio, affected the growth trend and dry matter accumulation per fruit. Maturation indices were not influenced by cane type. Therefore, a careful selection of renewal canes during winter pruning, within the larger diameter range used in the experiment, would enable an earlier harvest, with fruits characterized by optimal quality standards for the tested cultivar.

## Figures and Tables

**Figure 1 plants-14-02199-f001:**
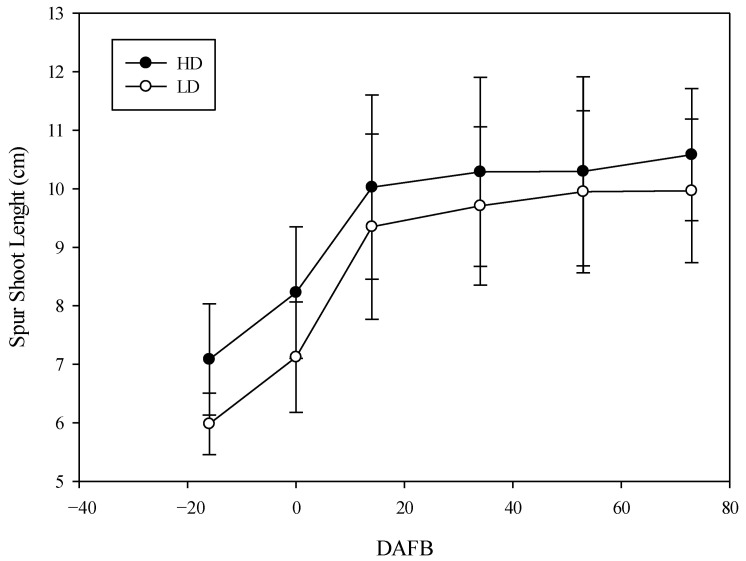
The development of spur (SP) length as a function of shoot type (average of two years). The whiskers represent the standard error [SE].

**Figure 2 plants-14-02199-f002:**
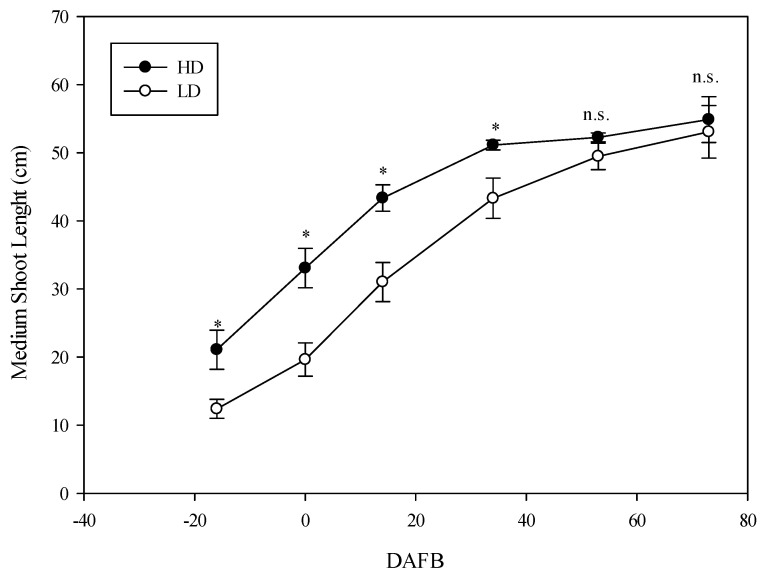
The development of the medium shoot (MS) length as a function of shoot type (average of two years). The whiskers represent the standard error [SE]. The asterisks show statistically significant differences (*p* < 0.05); n.s.: not significant.

**Figure 3 plants-14-02199-f003:**
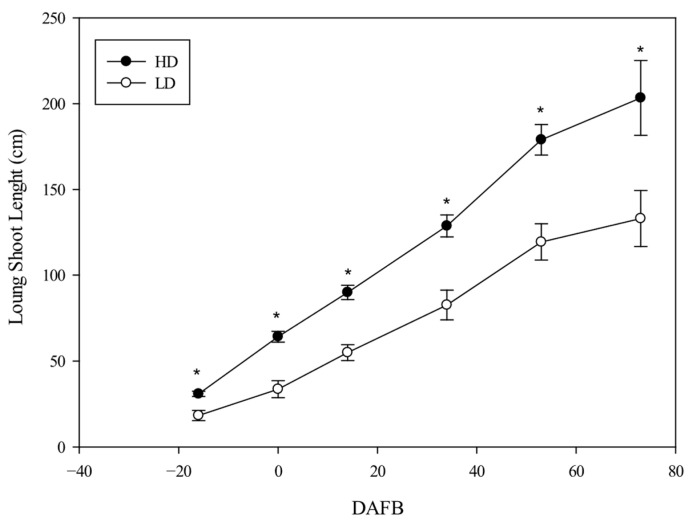
The development of long shoot (LS) length as a function of shoot type (average of two years). The whiskers represent the standard error [SE]. The asterisks show statistically significant differences (*p* < 0.05).

**Figure 4 plants-14-02199-f004:**
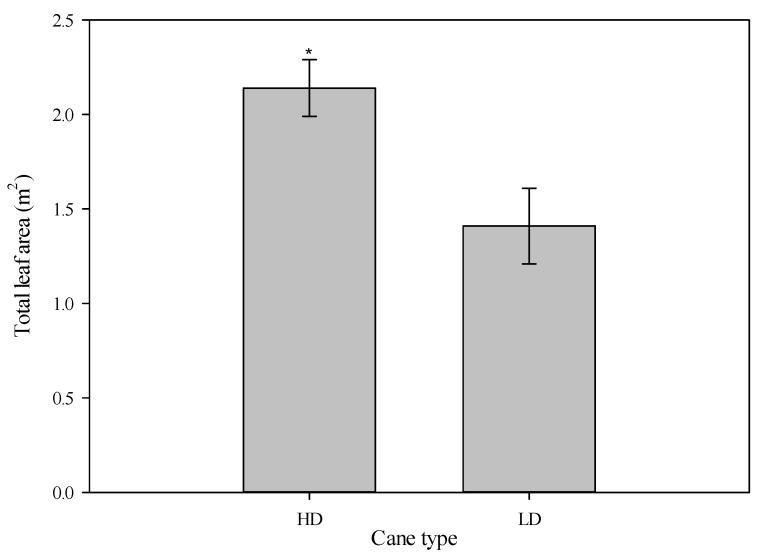
The total leaf area as a function of cane types (high-diameter (HD) and low-diameter (LD)), over an average of two years. The whiskers represent the standard error [SE]. The asterisk indicates significant differences, *p* ≤ 0.05.

**Figure 5 plants-14-02199-f005:**
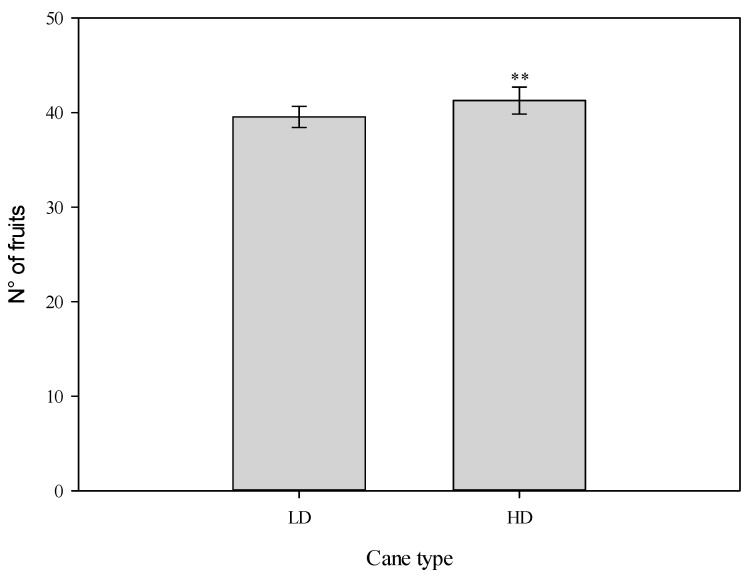
The number of fruits in relation to the type of cane in *Actinidia chinensis,* cv. Gold 3, high- diameter (HD) canes and low-diameter (LD) canes. The whiskers represent the standard error [SE]. The asterisks indicate significant differences, *p* ≤ 0.01.

**Figure 6 plants-14-02199-f006:**
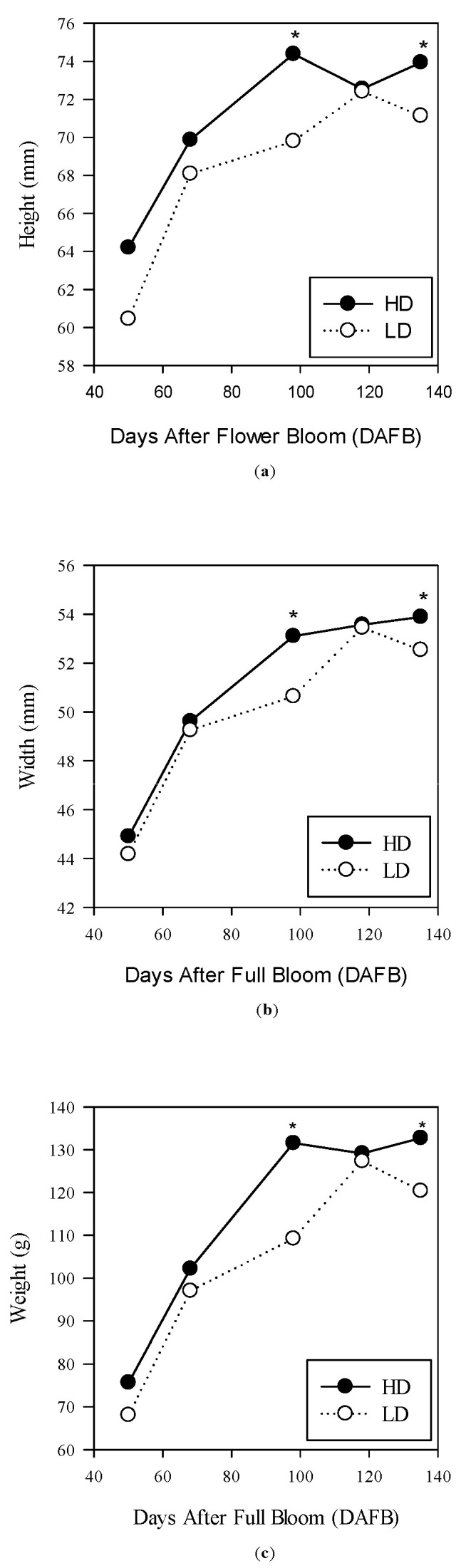
Evolutions of height (**a**), width (**b**), and fresh weight (**c**) in *A. chinensis* fruits, cv. Gold 3, based on cane type (high-diameter (HD) and low-diameter (LD)) over the two years of observation. The asterisks indicate significant differences at *p* ≤ 0.05.

**Figure 7 plants-14-02199-f007:**
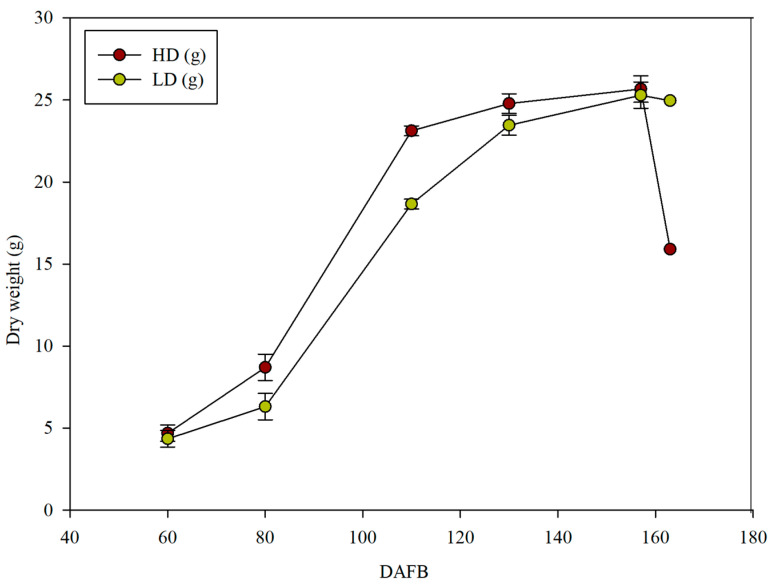
The evolution of dry weight in *Actinidia chinensis* fruits, cv. Gold 3, comparing the high-diameter (HD) shoot to the low-diameter (LD) shoot. The whiskers represent the standard error [SE].

**Figure 8 plants-14-02199-f008:**
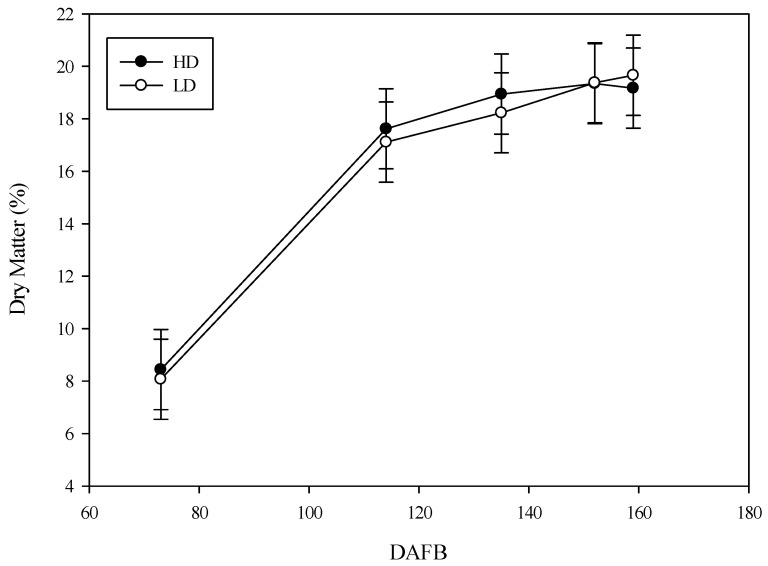
The difference in dry matter (%) in the fruit of *Actinidia chinensis*, cv. Gold 3, in the high-diameter (HD) and low-diameter (LD) shoots. The whiskers represent the standard error.

**Figure 9 plants-14-02199-f009:**
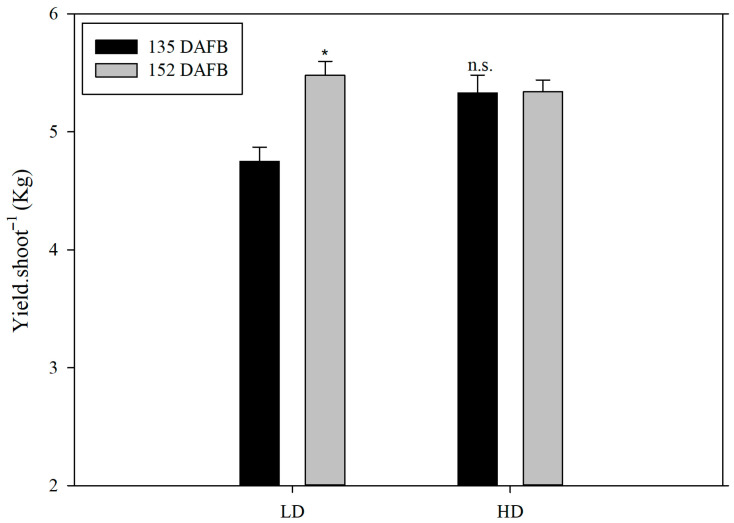
Production weights in the fruit of *A. chinensis*, cv. Gold 3, as a function of cane type (high-diameter (HD) and low-diameter (LD)). The whiskers represent the standard error [SE]. The asterisk indicates significant differences at *p* ≤ 0.05; n.s.: non-significant differences.

**Figure 10 plants-14-02199-f010:**
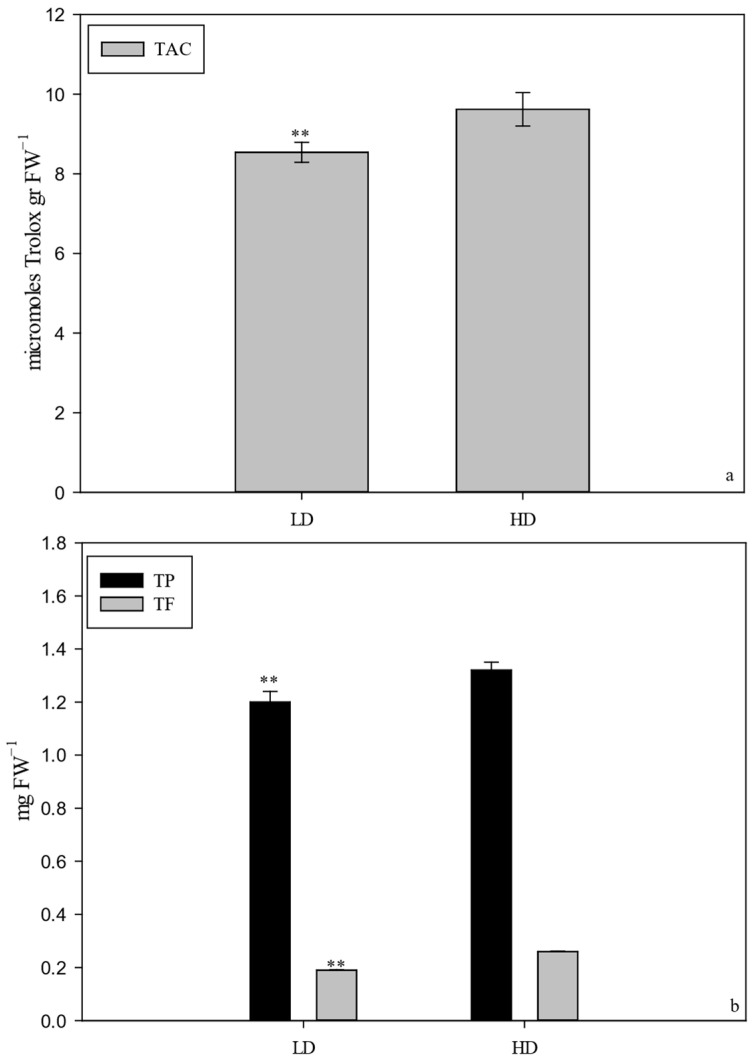
The main nutraceutical parameters (TAC: (**a**); TP: (**b**)) in the fruits of *Actinidia chinensis*, cv. Gold 3, at harvest, comparing the HD and LD treatments. The whiskers represent the standard error [SE]. The asterisks indicate significant differences at *p* ≤ 0.01.

**Figure 11 plants-14-02199-f011:**
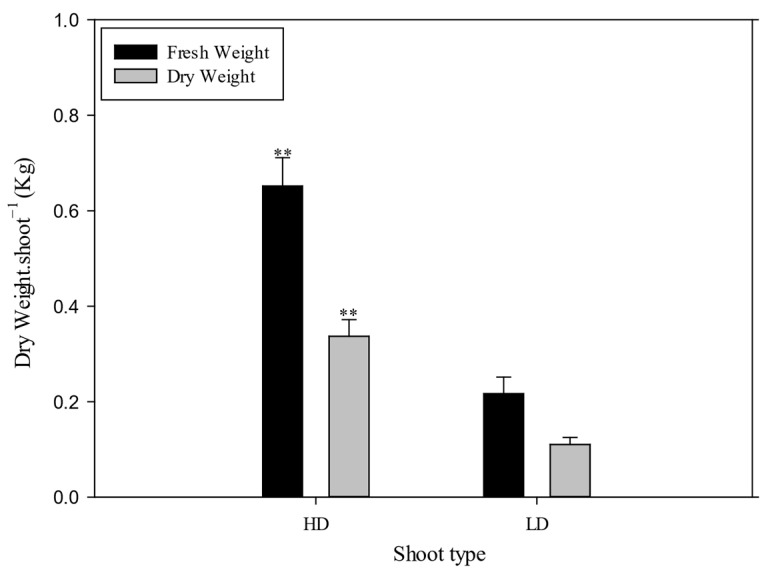
Fresh and dry weights for each shoot type (high-diameter (HD) and low-diameter (LD)), in plants of *Actinidia chinensis*, cv. Gold 3. The whiskers represent the standard error [SE]. The asterisks indicate significant differences at *p* ≤ 0.01.

**Table 1 plants-14-02199-t001:** Mean percentages of closed buds (CS), buds in the budburst stage (BBCH phase 1–10; BS), and in the vegetative stage (BBCH phase 10–18, VS) on canes categorized as HD (high-diameter) and LD (low-diameter) in *A. chinensis* plants, cv. Gold 3, over two years.

	Treatment (T)	Phenological Stage		
		CS	BS	VS
The first 10 d of April	LD	13.21 ± 1.98 n.s.	55.21 ± 5.50 n.s.	31.57 ± 6.8 n.s.
	HD	19.07 ± 2.03	55.97 ± 3.73	24.95 ± 3.95
The second 10 d of April	LD	14.83 ± 2.38 n.s.	31.07 ± 3.09 n.s.	46.82 ± 4.42 n.s.
	HD	19.17 ± 2.06	29.99 ± 2.48	50.82 ± 2.37
The first 10 d of May	LD	14.82 ± 6.31 n.s.	37.96 ± 3.23 n.s.	55.94 ± 2.94 n.s.
	HD	18.90 ± 1.56	30.89 ± 1.85	50.20 ± 2.20
Years (Y)		n.s.	n.s.	n.s.
Interaction Y × T		n.s.	n.s.	n.s.

n.s.: not significant.

**Table 2 plants-14-02199-t002:** Percentage of spur shoots (SP), medium shoots (MSs), and long shoots (LSs) on two cane types, with high (HD) and low (LD) diameters, in plants of *A. chinensis*, cv. Gold 3.

Treatment (T)	SP	Ms	Ls
LD	52.67 ± 4.13 **	10.43 ± 3.41 **	28.29 ± 5.26 **
HD	19.03 ± 6.48	19.03 ± 5.22	70.53 ± 5.16
Years (Y)	n.s.	n.s.	n.s.
T × Y	n.s.	n.s.	n.s.

** The asterisks in the column indicate significant differences at *p* ≤ 0.01; n.s.: not significant.

**Table 3 plants-14-02199-t003:** Number of leaves and leaf area by shoot type [HD (high-diameter) and LH (low-diameter)] in plants of *A. chinensis*, cv. Gold 3.

Treatment (T)	Leaves n°	Leaf Area cm^2^
SP-HD	6.0 ± 0.78 d	578.94 ± 8.75 d
SP-LD	5.8 ± 0.42 d	528.14 ± 7.35 d
MS-HD	13.75 ± 2.37 c	1326.74 ± 9.64 c
MS-LD	13.25 ± 1.15 c	1295.12 ± 8.95 c
LS-HD	26.5 ± 4.35 a	2573.06 ± 12.00 a
LS-LD	17.6 ± 1.35 b	1711.08 ± 4.35 b
Years (Y)	*	*
Treatment (T)	n.s.	n.s.
Y × T	n.s.	n.s.

Different letters indicate statistically significant differences at *p*-value < 0.05. * The asterisks in the column indicate significant differences, *p* ≤ 0.05; n.s.: not significant.

**Table 4 plants-14-02199-t004:** Principal ecophysiological parameters (Pn: photosynthesis net; gs: stomatal conductance; Ci: internal CO_2_ concentration; E: transpiration rate; LF: leaf temperature; LWP: leaf water potential) and fluorimetric indices (Fv’/Fm’: maximum quantum efficiency; PhiPS2: quantum efficiency of photosystem II; ETR: electron transport rate) of fruits in relation to the type of cane in *Actinidia chinensis* cv. Gold 3 (HD and LD shoots), averaged over two years.

Epoch	Treatment (T)	Pn µmol CO_2_ m^−2^ s^−1^	gs µmol H_2_O m^–2^ s^–1^	Ci ppm	Fv’/Fm’	PhiPS2	ETR	E mmol H_2_O m^−2^ s^−1^	LF °C	Leaf Water Potential BAR
June	HD	18.8 ± 0.11 *	0.23 ± 0.02 *	251.55 ± 2.15 *	0.62 ± 0.15 *	0.25 ± 0.01 *	157.14 ± 0.55 *	2.99 ± 0.15 *	30.53 ± 0.15 *	−4.04 ± 0.01 *
	LD	17.22 ± 0.12	0.20 ± 0.02	232.1 ± 0.01	0.57 ± 0.02	0.23 ± 0.23	132.15 ± 1.76	2.41 ± 0.23	31.49 ± 0.01	−6.22 ± 0.11
July	HD	18.6 ± 0.08 *	0.23 ± 0.01 *	248.61 ± 1.75 *	0.62 ± 0.11 *	0.23 ± 0.02 *	153.11 ± 0.34 *	2.95 ± 0.13 *	31.02 ± 0.11 *	−4.08 ± 0.00 *
	LD	16.84 ± 0.11	0.18 ± 0.01	231.22 ± 0.01	0.57 ± 0.02	0.19 ± 0.12	130.22 ± 1.15	2.22 ± 0.11	31.540 ± 0.04	−6.54 ± 0.13
August	HD	17.8 ± 0.06 *	0.21 ± 0.03 *	242.86 ± 1.88 *	0.59 ± 0.08 *	0.22 ± 0.03 *	152.18 ± 0.33 *	2.88 ± 0.11 *	31.52 ± 0.01 *	−4.23 ± 0.02 *
	LD	16.42 ± 0.09	0.17 ± 0.02	230.85 ± 0.03	0.55 ± 0.00b	0.18 ± 0.15	127.21 ± 1.18	2.04 ± 0.12	31.580 ± 0.06	−6.87 ± 0.15
Epoch		*	*	*	*	*	*	*	*	*.
Y (Year)		*	*	*	*	*	*	*	*	n.s.
Y × T		n.s.	n.s.	n.s.	n.s.	n.s.	n.s.	n.s.	n.s.	n.s.

The asterisks indicate significant differences at *p* ≤ 0.05. n.s.: not significant.

**Table 5 plants-14-02199-t005:** Number of fruits per shoot type as a function of cane diameter, (high-diameter (HD) and low-diameter (LD)) in plants of *A. chinensis*, cv. Gold 3.

Treatment	Shoot Type		
	SP	MSs	LSs
LD	2.33 ± 0.55 b	3.33 ± 0.61 a	3.71 ± 0.49 a
HD	2.27 ± 0.66 b	3.75 ± 0.20 a	3.57 ± 0.36 a
Year (Y)	n.s.		
Treatment	n.s.		
T × Y	n.s.		

Different letters in the row indicate differences at *p* ≤ 0.05 within shoot type; n.s.: non-significant differences.

**Table 6 plants-14-02199-t006:** Distribution of fruits (%) by size classes at harvest, based on the number of fruits produced per cane type (high-diameter (HD) and low-diameter (LD)) in *Actinidia chinensis* cv. Gold 3.

Shoot Type	Calibration Class				
	80–90	90–110	110–120	120–140	140–150
HD	28.57 *	28.57 *	22.86 *	14.28 *	5.714 *
LD	22.85	57.14	14.28	5.714	2.857
year	n.s.	n.s.	n.s.	*	*

The asterisks in the column indicate significant differences, *p* ≤ 0.05; n.s.: non-significant differences.

**Table 7 plants-14-02199-t007:** The number of leaves per fruit and leaf area per fruit by shoot type, as a function of cane diameter [HD (high-diameter) and LD (low-diameter)] in *A. chinensis* plants, cv. Gold 3.

Treatment	Number Leaves n°	Leaf Area cm^2^
SP	3.09 ± 0.77 c	2.49 ± 0.40 c
MS	5.50 ± 1.69 b	3.81 ± 0.39 b
LS	9.47 ± 1.93 a	8.56 ± 1.19 a

Different letters in the column indicate differences that are significant at *p* ≤ 0.05.

**Table 8 plants-14-02199-t008:** Trends in the main ripening indices (TSSs, total soluble solids; Fir, firmness; TA: titratable acidity) in fruits of *Actinidia chinensis*, cv. Gold 3, at harvest, comparing the HD and LD treatments.

DAFB	Treatment	TSS (°Brix)	Fir (Kg.cm^−2^)	TA (%)	TSS/TA
73	HD	6.00 ± 0.18 n.s.	10.17 ± 0.17 n.s.	1.40 ± 0.03 n.s.	2.27 ± 0.02 n.s.
	LD	5.96 ± 0.15	10.19 ± 0.19	1.47 ± 0.05	2.08 ± 0.02
112	HD	6.50 ± 0.31 n.s.	9.31 ± 0.18 n.s.	1.58 ± 0.03 n.s.	1.99 ± 0.01 n.s.
	LD	6.49 ± 0.28	8.79 ± 0.19	1.64 ± 0.03	1.99 ± 0.02
133	HD	7.26 ± 0.15 n.s.	9.08 ± 0.21 n.s.	1.58 ± 0.05 n.s.	2.01 ± 0.03 n.s.
	LD	7.58 ± 0.18	8.44 ± 0.20	1.65 ± 0.02	1.86 ± 0.01
155	HD	9.64 ± 0.15 n.s.	8.10 ± 0.15 n.s.	1.58 ± 0.02 n.s.	2.04 ± 0.02 n.s.
	LD	10.44 ± 0.18	8.34 ± 0.18	1.55 ± 0.02	2.34 ± 0.03

n.s.: non-significant differences. Trends of the main ripening indices (TSS, total soluble solids; Fir, firmness; TA, titratable acidity) in fruits of *Actinidia chinensis*, cv. Gold 3, at harvest, comparing the HD and LD treatments.

**Table 9 plants-14-02199-t009:** Main colorimetric parameters (L*: lightness; a* and b*: chromatic coordinates; Ch: chroma; °hue: tint) in the fruits of *Actinidia chinensis*, cv. Gold 3, at harvest, comparing the HD and LD treatments.

DAFB	Treatment	L*	a*	b*	Ch	°Hue
73	LD	64.4 ± 0.52 n.s.	−6.10 ± 0.61 n.s.	24.4 ± 0.22 n.s.	25.2 ± 0.41 n.s.	103.95 ± 0.20 n.s.
	HD	64.8 ± 0.31	−6.18 ± 0.55	24.9 ± 0.21	25.6 ± 0.45	103.88 ± 0.19
112	LD	75.3 ± 0.30 n.s.	−4.52 ± 0.48 n.s.	21.0 ± 0.21 n.s.	21.7 ± 0.21 n.s.	101.84 ± 0.22 n.s.
	HD	73.6 ± 0.23	−3.71 ± 0.46	19.9 ± 0.26	20.1 ± 0.22	100.51 ± 0.23
133	LD	75.1 ± 0.40 n.s.	−1.73 ± 0.48 n.s.	16.73 ± 0.31 n.s.	16.7 ± 0.51 n.s.	94.4 ± 0.35 n.s.
	HD	74.4 ± 0.28	−2.27 ± 0.35	16.62 ± 0.34	12.9 ± 0.55	94.23 ± 0.26
155	LD	74.8 ± 0.25 n.s.	−0.04 ± 0.18 n.s.	14.7 ± 0.35 n.s.	14.8 ± 0.22 n.s.	91.38 ± 0.28 n.s.
	HD	74.4 ± 0.22	−0.17 ± 0.21	14.8 ± 0.36	14.9 ± 0.28	91.76 ± 0.21

n.s.: non-significant differences.

## Data Availability

The original contributions presented in this study are included in the article. Further inquiries can be directed to the corresponding author.

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
