# Peer review of "Differences in Vegetative, Productive, and Physiological Behaviors in Actinidia chinensis Plants, cv. Gold 3, as A Function of Cane Type"

_plants, 2025, doi:10.3390/plants14142199_

Round 1
Reviewer 1 Report
Comments and Suggestions for Authors
The manuscript presents a comprehensive investigation into the effects of cane diameter on vegetative growth, physiological responses, and fruit quality in Actinidia chinensis cv. Gold 3. The study is well-designed, with clear objectives and robust methodologies. However, several areas require clarification to enhance the manuscript’s impact and readability. Below are detailed comments and suggestions for improvement.
- Define “acrotonic tendency”and “preformed vs. neoformed phytomers” for readers unfamiliar with kiwifruit morphology.
- Clarify why a two-way ANOVA was used for some parameters while a t-test was applied to others. Justify the choice of tests.
- Include degrees of freedom and F-values for ANOVA results in tables.
- Provide climatic data for the experimental site, as these could influence physiological response.
- Specify how fruits were selected for destructive sampling to ensure representativeness.
- In Figures 1–3, include error bars or confidence intervals to visualize variability.
- Figure 6: Label axes more clearly. For example, Days After Full Bloom.
- Use consistent units. For example, “cm²” for leaf area.
- Simplify column headers.
- Expand on why HD canes had higher hydraulic conductance. Is this due to xylem vessel size, number, or other anatomical traits?
- Discuss whether higher photosynthesis in HD canes is driven by stomatal or biochemical factor.
Minor Comments:
Replace “theses” with “treatments”.
“Fv/Fm+” (Table 4) should be “Fv/Fm.”
Define all abbreviations at first use.
Author Response
Dear reviewer, thank you for reviewing the manuscript. We welcome your comments and suggestions. We hope we have addressed the comments satisfactorily and that the manuscript is now acceptable for publication.
- Define “acrotonic tendency”and “preformed vs. neoformed phytomers” for readers unfamiliar with kiwifruit morphology.
The suggestion was accepted. Lines 52-55.
- Clarify why a two-way ANOVA was used for some parameters while a t-test was applied to others. Justify the choice of tests.
The reference was reported in the materials and methods section: statistical analysis. Lines 291-293
- Include degrees of freedom and F-values for ANOVA results in tables.
The degrees of freedom and F-values have been inserted in the tables attached as supplementary files for the main kiwifruit quality indices.
- Provide climatic data for the experimental site, as these could influence physiological response.
The suggestion was accepted. The thermopuviometric data have been included as supplementary files (Figure S1). The cold units for the two years of testing have been reported in the text. See lines 106-109.
- Specify how fruits were selected for destructive sampling to ensure representativeness.
The suggestion was accepted. Lines 135-136.
- In Figures 1–3, include error bars or confidence intervals to visualize variability.
The suggestion was accepted
- Figure 6: Label axes more clearly. For example, Days After Full Bloom.
The suggestion was accepted
- Use consistent units. For example, “cm²” for leaf area.
The suggestion was accepted
- Simplify column headers.
The suggestion was accepted
- Expand on why HD canes had higher hydraulic conductance. Is this due to xylem vessel size, number, or other anatomical traits?
The integration has been made. Lines 580-581
- Discuss whether higher photosynthesis in HD canes is driven by stomatal or biochemical factor.
Lines 590-593.
- Replace “theses” with “treatments”.
The suggestion was accepted
- “Fv/Fm+” (Table 4) should be “Fv/Fm.”
The suggestion was accepted
- Define all abbreviations at first use.
The suggestion was accepted

Reviewer 2 Report
Comments and Suggestions for Authors'Differences in vegetative, productive, and physiological behaviors in Actinidia chinensis plants, cv. Gold 3, as a function of cane type' is a very well written manuscript especially considering the quality of presentation of the results and discussion.
1) However, to compensate comparatively lower novelty, authors could strengthen the justification of this study by further highlighting the importance of vegetative growth on final fruit yield and vine sustainability under the introduction section.
2) In lines 71, 79, 85, 94, and 95 please italicise the word Actinidia
3) In the materials and methods and results sections (in whole paper) please replace the word decade with 10 d.
4) In materials and methods, results and discussion sections, please use SI units for days (d), hours (h), minutes (min), seconds (s) and keep consistency with the units e.g in some sections the units were given as per g in others /g etc. Overall, the units lack consistency.
5) Under results, the authors have stated (lines 358,359) that the stomatal conductance was significantly lower in HD canes, but in the discussion section they mentioned (line 534) higher stomatal conduction in HD canes. I guess there is a typo in the results section (lines 358, 3589). Results table 4 shows significantly higher values for HD canes.
6) Under the discussion section: If available, try to find any reported studies comparing the levels of cytokinins and gibberellins in the shoots/ canes of different thicknesses. That too can be related to the better fruit size/ weight.
7) It is recommended but not compulsory to carry out a co-relation analysis (e.g. R software, Corrplot) to strengthen the arguments made in the discussion (line 518).
Author Response
Dear reviewer, thank you for reviewing the manuscript. We welcome your comments and suggestions. We hope we have addressed the comments satisfactorily and that the manuscript is now acceptable for publication
1) However, to compensate comparatively lower novelty, authors could strengthen the justification of this study by further highlighting the importance of vegetative growth on final fruit yield and vine sustainability under the introduction section.
The suggestion was accepted and the introduction was integrated
2) In lines 71, 79, 85, 94, and 95 please italicise the word Actinidia
The correction has been made
3) In the materials and methods and results sections (in whole paper) please replace the word decade with 10 d.
The correction has been made
4) In materials and methods, results and discussion sections, please use SI units for days (d), hours (h), minutes (min), seconds (s) and keep consistency with the units e.g in some sections the units were given as per g in others /g etc. Overall, the units lack consistency.
The correction has been made
5) Under results, the authors have stated (lines 358,359) that the stomatal conductance was significantly lower in HD canes, but in the discussion section they mentioned (line 534) higher stomatal conduction in HD canes. I guess there is a typo in the results section (lines 358, 3589). Results table 4 shows significantly higher values for HD canes.
Yes, it was a typing error. The correction has been made.
6) Under the discussion section: If available, try to find any reported studies comparing the levels of cytokinins and gibberellins in the shoots/ canes of different thicknesses. That too can be related to the better fruit size/ weight.
In this study, we do not have data on hormonal relationships, but a future experiment is being considered to investigate hormonal interactions in relation to cane size.

Reviewer 3 Report
Comments and Suggestions for Authors
Dear Authors,
I have read your highly relevant experiment concerning kiwi cultivation and the relationship between cane size and overall plant performance and yield. I found your results interesting from a practical standpoint and worth disseminating. The experiments were well-designed, and the methods are described thoroughly. The visual presentation of the results is generally clear, though there are a few minor issues that require correction. Overall, I recommend your article for publication after minor revisions and corrections.
Comments and suggestions:
I suggest considering a more engaging and informative title.
I am not sure whether the term "evolution," used to describe differences in growth, development, and morphological changes, is the most appropriate in this context. It may be misunderstood, also by searching engines. Please, consider using "development" and "alternations" in this context.
When describing the methodology for fruit assessment (e.g., lines 205, 215), please clarify that it refers specifically to the fruit.
Tables and graphs:
Please explain all abbreviations (e.g., TAC, TP, TF, DAFP) in the captions so that they are self-explanatory, without the need to refer back to the methodology section.
For graphs, clarify what the whiskers represent (e.g., standard deviation [SD] or standard error [SE]).
In Table 4 and others: do the letters (a, b, etc.) refer to comparisons within a single treatment (LD or HD), or across both? Please indicate clearly. If they refer to separate treatments, consider using lowercase and uppercase letters within the same column.
Be consistent with the formatting of species names in captions; I recommend italicizing A. chinensis throughout.
In Table 3, use the letter "a" for the highest value, "b" for the second highest, and so on, to maintain clarity.
Below tables and graphs that present statistical results, indicate the type of statistical test used.
The discussion is missing a section on how cane size affects fruit quality, as well as the physiological status of the plants. Please include a discussion that refers to Fv/Fm values in this context.
Line 568: Please verify if the term "long-diameter" is correct and appropriate in that context.
Best regards,
Author Response
Dear reviewer, thank you for reviewing the manuscript. We welcome your comments and suggestions. We hope we have addressed the comments satisfactorily and that the manuscript is now acceptable for publication.
- When describing the methodology for fruit assessment (e.g., lines 205, 215), please clarify that it refers specifically to the fruit.
We included the term fruit in the paragraph title and added the methodology related to nutraceutical aspects as a sub-paragraph
- I am not sure whether the term "evolution," used to describe differences in growth, development, and morphological changes, is the most appropriate in this context. It may be misunderstood, also by searching engines. Please, consider using "development" and "alternations" in this context.
The changes have been made, keeping the term evolution in reference to phenological stages and in other instances where its use is appropriate.
- Please explain all abbreviations (e.g., TAC, TP, TF, DAFP) in the captions so that they are self-explanatory, without the need to refer back to the methodology section.
The suggestion was accepted and the abbreviations are explained in the captions of figures and tables.
- For graphs, clarify what the whiskers represent (e.g., standard deviation [SD] or standard error [SE]).
The suggestion was accepted and the information was reported in the caption of the graphs.
- In Table 4 and others: do the letters (a, b, etc.) refer to comparisons within a single treatment (LD or HD), or across both? Please indicate clearly. If they refer to separate treatments, consider using lowercase and uppercase letters within the same column.
The suggestion was accepted.
- Be consistent with the formatting of species names in captions; I recommend italicizing A. chinensis throughout.
The correction has been made.
- In Table 3, use the letter "a" for the highest value, "b" for the second highest, and so on, to maintain clarity.
The correction has been made.
- Below tables and graphs that present statistical results, indicate the type of statistical test used.
The reference was reported in the materials and methods section: statistical analysis. Lines 291-293
- The discussion is missing a section on how cane size affects fruit quality, as well as the physiological status of the plants. Please include a discussion that refers to Fv/Fm values in this context.
The reference has now been included in lines 590–593, as indicated.
- Line 568: Please verify if the term "long-diameter" is correct and appropriate in that context.
The correction has been made. Lines 616-618.

Round 2
Reviewer 1 Report
Comments and Suggestions for Authors
The author has revised the reviewer's opinion very well.